# PHYSICALLY GROUNDED AVATAR GENERATION

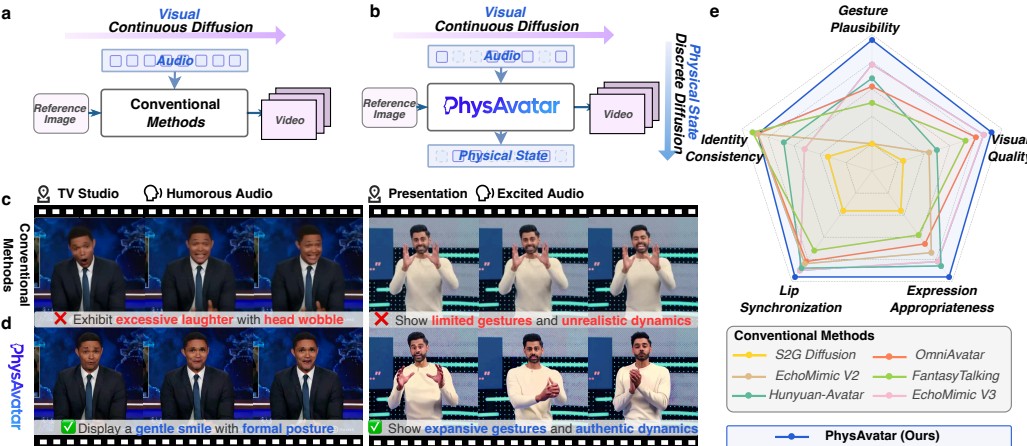

Figure 1: **Training schemes of both conventional methods and our PhysAvatar with qualitative examples and visual Turing radar results.** (a-b) Training schemes for conventional and our PhysAvatar. (c-d) Comparative qualitative examples. (e) Visual Turing radar results.

## ABSTRACT

Recent advances in diffusion transformer (DiT) models have greatly improved audio-driven video avatar generation, enabling the synthesis of realistic avatars from a single reference image and an audio clip. However, generating avatars with *physically grounded human behaviors* remains challenging, primarily due to (**i**) overreliance on shallow audio-visual correlations and (**ii**) misalignment between semantic intent and behavioral expression. Consequently, existing methods often produce facial expressions and gestures that appear constrained, lack emotional depth, and fail to capture realistic human dynamics. In this paper, we present a **Phys**ically grounded DiT model for **Avatar** generation, termed **PhysAvatar**, which can produce realistic, contextually coherent, long-form avatars with human-like behavioral fidelity. PhysAvatar introduces three key innovations: (**i**) physical state supervision, embedding human behavioral dynamics into the video DiT model via discrete diffusion; (**ii**) physical planning guidance, which leverages a multimodal language model to jointly analyze audio and visual inputs and direct the avatar behaviors according to semantic intent; and (**iii**) efficient long-form inference with interleaved video interpolation, improving temporal coherence and identity preservation. Extensive experiments on our in-house dataset, as well as PATS and Vlogger, demonstrate that PhysAvatar outperforms state-of-the-art baselines in both generative quality and behavioral realism, consistently producing avatars that are more physically grounded, expressive, and lifelike.

## 1 INTRODUCTION

Audio-driven video avatar generation (Meng et al., 2025b; Chen et al., 2025; Wang et al., 2025; Meng et al., 2025a; Gan et al., 2025) aims to synthesize lifelike avatar videos from a single reference image and an audio clip by animating lip movements, facial expressions, and body gestures based on audio. This capability is increasingly important in entertainment, education, and interactive media, where users demand not only precise audio-visual synchronization but also physically plausible behaviors.

Recent advances in video synthesis have been propelled by Diffusion Transformer (DiT) models, such as Sora (Peebles & Xie, 2023) and Wan (Wan et al., 2025). However, state-of-the-art (SOTA)

DiT-based audio-driven avatar methods (Figure 1(**a**)) primarily capture low-level audio-visual corre-lations, *e.g.*, phoneme-to-lip mappings or prosody-to-gesture intensity, often overlooking whether the generated movements reflect *physically grounded human behavior*. Consequently, while lip synchronization may be accurate, facial expressions often lack emotional depth, gestures appear "floaty", and dynamic movements may misalign with the intended context (Figure 1(**c**)).

We identify two critical gaps in current methods. First, **overreliance on audio-visual correlations**: existing methods largely depend on conventional visual continuous diffusion supervision (Ho et al., 2020; Liu et al., 2023), which oversimplifies the audio-motion relationship. This leads to artifacts like jitter, unnatural gestures, and desynchronization between facial and body movements. Second, **semantic intent misalignment**: natural motion should reflect not only audio content and prosody but also identity cues and contextual factors from the reference image. Without this guidance, movements may be locally plausible yet semantically inconsistent. Addressing these gaps is essential for generating avatars that are both natural and human-like.

To address these challenges, we propose PhysAvatar, a **Physi**cally grounded DiT model for **Avatar** generation that produces realistic, contextually coherent, long-form avatars with human-like behav-ioral fidelity. Our PhysAvatar improves upon conventional video DiT-based avatar methods in two major aspects. **(i)** We introduce **physical state supervision** via a discrete diffusion mechanism, which embeds human behavioral dynamics into the intermediate video DiT layer (Figure 1(**b**)). By acknowledging that individuals express the same auditory content through diverse behaviors, we employ a discrete temporal masking strategy to predict masked physical state tokens from un-masked audio tokens, effectively capturing the behavioral nuances that emerge as audio evolves. The supervised physical state tokens are derived from the SOTA pose estimator X-Pose (Yang et al., 2024), which captures critical information regarding body, facial, and hand movements. **(ii)** We also provide **physical planning guidance** via a multimodal large language model (MLLM)-based guider, *i.e.* Qwen2.5-Omni (Xu et al., 2025), to analyze audio and images jointly, thereby capturing emotions and intentions while planning future behaviors. The MLLM integration equips the video DiT with state-transition guidance, ensuring avatar behaviors evolve consistently with the intended semantic context. In addition, we propose an efficient long-form inference strategy utilizing interleaved video interpolation to address behavioral inertia across sequential video chunks, effectively resolving the identity drifting issue. Collectively, these advancements produce avatars that are well-synchronized with audio and exhibit enhanced physical grounding, expressiveness, and lifelikeness (Figures 1(**d-e**)).

The contributions of our PhysAvatar are summarized as follows. **(i)** We present a novel physically grounded DiT model for avatar generation, which can produce realistic, contextually coherent, and long-form avatars with human-like behavioral fidelity. **(ii)** We propose a physical state supervision via discrete diffusion mechanism to embed human behavioral dynamics into the video DiT model. **(iii)** We utilize an MLLM-based guider to analyze audio and image inputs jointly and direct the avatar behaviors according to semantic intent. **(iv)** We develop an efficient long-form inference strategy with interleaved video interpolation, improving temporal coherence and identity preservation. **(v)** Experimental results on our in-house, PATS, and Vlogger datasets demonstrate that our PhysAvatar outperforms existing methods in generative performance, while also producing human behaviors that are more physically grounded, expressive, and lifelike.

## 2 RELATED WORKS

**Audio-driven video avatar generation.** Audio-driven video avatar generation aims to synthesize realistic avatars from a single reference image and an audio clip (Corona et al., 2025). Existing methods fall into the following two categories. 1) Explicit pose-intermediated models (He et al., 2024; Corona et al., 2025; Meng et al., 2025b), *i.e.* audio to pose to video, which maps audio to intermediate pose visualizations (*e.g.* 2D/3D keypoints) via an audio-to-pose model before video generation; however, they are susceptible to pose noise, latency, and audio-pose misalignment. 2) End-to-end models (Chen et al., 2025; Wang et al., 2025; Gan et al., 2025; Meng et al., 2025a), *i.e.* audio to video, which generate video directly from audio but often yield conservative, limited-amplitude, or semantically implausible facial and gesture motions. Despite notable gains in visual fidelity and synchronization, they both struggle to deliver physically grounded, human-like behavior. Unlike them, the proposed PhysAvatar is able to produce physically grounded avatars via physical state supervision and physical planning guidance.

**Video DiT model.** The video DiT model, exemplified by text-to-video (T2V) models such as Sora (Peebles & Xie, 2023; Brooks et al., 2024) and Wan-series (Wan et al., 2025), integrates

scalable transformers with visual continuous diffusion to jointly model spatial fidelity and temporal coherence. This architecture is currently the main thrust for audio-driven video avatar generation, showcasing remarkable abilities in capturing intricate interactions, motion, and environmental context. Additionally, frameworks extending from T2V foundation models, *e.g.* the All-in-One Video Creation and Editing (VACE) model (Jiang et al., 2025) from Wan2.1, facilitate a variety of downstream applications. However, these DiT-based models typically depend on visual continuous diffusion loss supervision, which prioritizes pixel-level fidelity but offers limited guidance for modeling human behavior. Thus, achieving physically grounded human behaviors remains challenging.

## 3 METHODOLOGY

In this section, we begin with the preliminaries in Section 3.1, followed by an overview of PhysAvatar in Section 3.2. Section 3.3 then introduces our discrete diffusion-based physical state supervision, while Section 3.4 describes the concrete MLLM-based physical planning guidance. Finally, Section 3.5 specifies the overall objective function together with our long-form inference strategy.

### 3.1 PRELIMINARIES

Diffusion models have emerged as a powerful framework in both computer vision (CV) and natural language processing (NLP). They operate via a two-phase procedure: a forward process that gradually corrupts clean data and a learned reverse process that reconstructs the original signal. Accordingly, they are commonly categorized as (**i**) *continuous diffusion*, which applies continuous-valued noise corruption (Ho et al., 2020), and (**ii**) *discrete diffusion*, which uses discrete masking (Lou et al., 2024).

**Continuous diffusion model.** We reinterpret conventional continuous-valued image and video diffusion in CV as continuous diffusion: a forward process that progressively perturbs continuous-valued data with Gaussian noise and a reverse process that predicts the noise (or velocity) required to recover the clean signal. Exemplified by latent-based rectified flows (Liu et al., 2023), let a pretrained encoder $\mathcal{E}(\cdot)$ map an image or video $\boldsymbol{X} \in \mathbb{R}^{F \times H \times W \times 3}$ to latents $\boldsymbol{x}_1 = \mathcal{E}(\boldsymbol{X}) \in \mathbb{R}^{(f \times h \times w) \times C}$, where $F$, $H$, and $W$ denote the frame number, height, and width of the original data, and $f$, $h$, $w$, and $C$ denote the corresponding latent dimensions. The forward process randomly samples Gaussian noise $\boldsymbol{x}_0 \sim \mathcal{N}(\boldsymbol{0}, \boldsymbol{1})$ and defines intermediate latents along a linear path $\boldsymbol{x}_t = t\boldsymbol{x}_1 + (1 - t)\boldsymbol{x}_0$ for the timestep $t \in [0, 1]$. In the reverse process, the diffusion model $\mathcal{U}_{\boldsymbol{\theta}}(\cdot)$ predicts the velocity by minimizing a mean-squared-error objective between its output and the ground-truth velocity $\boldsymbol{v}_t$:

$$\mathcal{L}_{\text{Continuous}} = \|\mathcal{U}_{\boldsymbol{\theta}}(\boldsymbol{x}_t, \boldsymbol{e}^{\text{c}}, t) - \boldsymbol{v}_t\|^2, \tag{1}$$

where $\boldsymbol{v}_t = \frac{d\boldsymbol{x}_t}{dt} = \boldsymbol{x}_1 - \boldsymbol{x}_0$ and $\boldsymbol{e}^{\text{c}}$ denotes the conditioning tokens.

**Discrete diffusion model.** Discrete diffusion is an efficient and powerful paradigm for NLP that formulates generation as progressive token masking followed by reconstruction. Exemplified by masked diffusion (Nie et al., 2025), given a sequence of text token embeddings $\boldsymbol{x}_0 \in \mathbb{R}^{L^{\text{txt}} \times C^{\text{txt}}}$, where $L^{\text{txt}}$ is the sequence length and $C^{\text{txt}}$ is the embedding dimensionality, the forward process yields a partially observed sequence $\boldsymbol{x}_t \in \mathbb{R}^{L^{\text{txt}} \times C^{\text{txt}}}$ by independently replacing each token $\boldsymbol{x}_t^i$ with a dedicated mask embedding $\boldsymbol{e}^{\text{m}} \in \mathbb{R}^{C^{\text{txt}}}$ with probability $t \in [0, 1]$. The reverse process then trains a model $p_{\boldsymbol{\theta}}(\cdot)$ to recover the original tokens at masked positions, conditioned on the visible context $\boldsymbol{x}_t$. Accordingly, the learning objective is a masked negative log-likelihood:

$$\mathcal{L}_{\text{Discrete}} = -\mathbf{1}[\boldsymbol{x}_t^i = \boldsymbol{e}^{\text{m}}] \log p_{\boldsymbol{\theta}}(\boldsymbol{x}_0^i \mid \boldsymbol{x}_t), \tag{2}$$

where $\mathbf{1}[\cdot]$ is the indicator function that restricts supervision to masked tokens.

### 3.2 OVERVIEW OF PHYSAVATAR

Given a reference image of a human and an audio clip, PhysAvatar aims to produce a realistic avatar that effectively aligns lip movement, expression, and gesture with the audio's acoustic and linguistic content, along with the physical environment depicted in the image. As shown in Figure 2, we adopt the T2V model Wan2.1 (Wan et al., 2025) in the *main branch* enhanced with a dedicated *VACE branch*, which builds on the advanced VACE model (Jiang et al., 2025) and the principles of

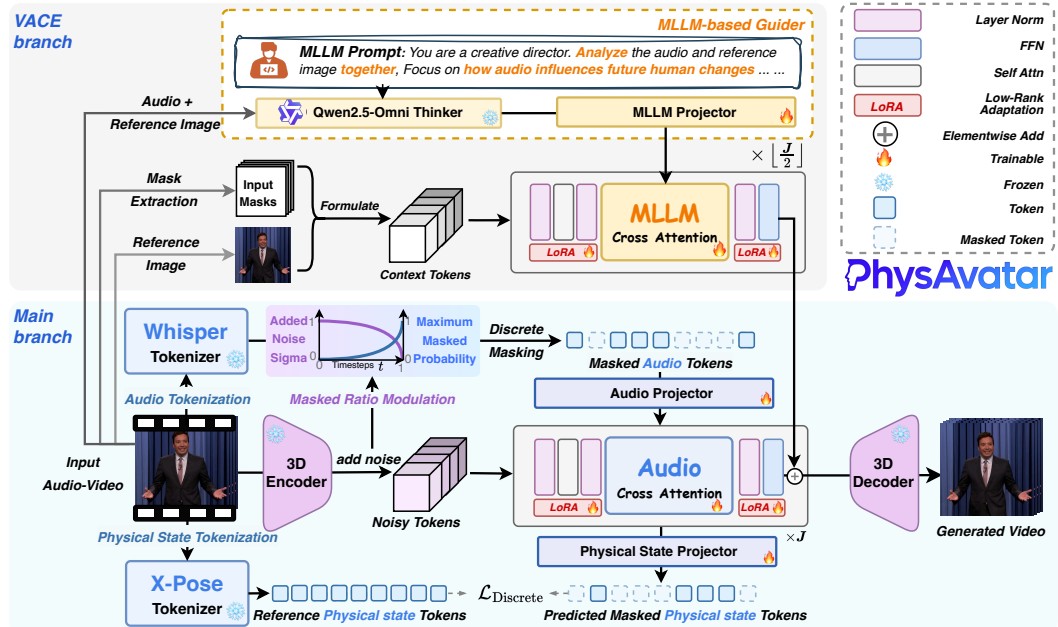

Figure 2: **Illustration of the training process for the proposed PhysAvatar.**

ReferenceNet (Hu, 2024). Specifically, we inject audio tokens into the main branch to modulate fine-grained dynamics, whereas the reference image and MLLM embeddings are injected into the VACE branch with additional transformer blocks to provide high-level physical planning guidance.

During training, we optimize the video DiT model using (i) discrete diffusion-based physical state supervision and (ii) continuous diffusion-based visual supervision, resulting in videos that exhibit more realistic human dynamics and enhanced visual quality. The inference process remains consistent with the standard video DiT model, without masking any audio tokens. Additionally, we propose an efficient long-form inference strategy utilizing interleaved video interpolation, enabling the generation of temporally coherent and identity-preserving long avatar videos.

## 3.3 DISCRETE DIFFUSION-BASED PHYSICAL STATE SUPERVISION

To promote physically grounded human behavior in conventional video DiT models, we propose a discrete diffusion-based physical state supervision, thereby facilitating the learning of temporal behavior changes as audio progresses, as shown in the main branch of Figure 2.

**Audio tokens injection.** We condition video generation on input audio by injecting Whisper (Radford et al., 2023)-tokenized audio tokens into each transformer block of the main branch via cross-attention. To expand the audio receptive field and enhance temporal continuity, we replace the raw encoded audio tokens $e^{\text{raw\_aud}} \in \mathbb{R}^{F \times C^{\text{aud}}}$ with overlapping sliding-window tokens $e^{\text{win\_aud}} \in \mathbb{R}^{F \times \left((2k+1) \times C^{\text{aud}}\right)}$, using a window size of $2k + 1$:

$$e^{\text{win\_aud}} = \oplus_{i=1}^{F} [e_{i-k}^{\text{raw\_aud}}, \dots, e_i^{\text{raw\_aud}}, \dots, e_{i+k}^{\text{raw\_aud}}], \tag{3}$$

where the superscript $^{\text{aud}}$ denotes audio, the subscript $i$ refers to the $i$-th token, $[\cdot, \cdot]$ denotes row-wise (horizontal) concatenation, and $\oplus$ denotes column-wise (vertical) concatenation, respectively; mirror padding is applied at sequence boundaries. Next, we introduce an MLP-based audio projector, $\text{Proj}_{\text{aud}}(\cdot)$, which temporally compresses $e^{\text{win\_aud}}$ to produce the final audio tokens $e^{\text{aud}} = \text{Proj}_{\text{aud}}(e^{\text{win\_aud}}) \in \mathbb{R}^{f \times C}$. This ensures alignment with the video latent sequence length $f$ while maintaining a dimensionality of $C$. Additionally, rotary positional encoding (RoPE) (Su et al., 2024) is incorporated into the audio cross-attention to facilitate audio-visual temporal alignment.

**Noise-based audio masked ratio modulation.** In continuous diffusion, the timestep noise $\sigma_t$ decays from global to local learning; naive discrete supervision breaks this curriculum, causing scale mismatch and unstable gradients. We therefore propose a noise-based audio masked ratio modulation strategy to facilitate the alignment of both objectives per timestep, as detailed in Alg. 1.

---

**Algorithm 1:** Noise-based Audio Masked Ratio Modulation

---

**Require**: video latent sequence length $f$, current training sampled timestep $t$
**Output**: discrete audio mask $\boldsymbol{M}^{\text{aud}} \in \{0,1\}^f$
// Compute maximum masked probability
**Get** added $t$-th noise sigma $\sigma_t \in [0,1]$ via noise scheduler;
**Compute** maximum masked probability $P_t^{\text{M}} = 1 - \sigma_t$;
// Compute the number of masked audio tokens
**Sample** a masking ratio $p^{\text{M}} \sim \mathcal{U}(0, P_t^{\text{M}})$;
**Compute** the number of masked audio tokens $n^{\text{M}} = \lfloor f \times p^{\text{M}} \rfloor$;
// Generate the discrete mask for physical state supervision
**Select** mask indices $\mathbb{I}^{\text{M}} \subseteq \{1, \ldots, f\}$, randomly and uniquely, where $|\mathbb{I}^{\text{M}}| = n^{\text{M}}$;
**Return** $\boldsymbol{M}^{\text{aud}}$ where $\boldsymbol{M}^{\text{aud}}[i] = \begin{cases} 0 & \text{if } i \in \mathbb{I}^{\text{M}} \\ 1 & \text{if } i \notin \mathbb{I}^{\text{M}} \end{cases}$ for $i = 1, 2, \ldots, f$

---

Specifically, at a given timestep $t$ with a continuous noise scale $\sigma_t$, we first establish the maximum audio-masking probability $P_t^{\text{M}}$ based on $\sigma_t$, linking the discrete and continuous diffusion processes. We then sample the masking ratio $p^{\text{M}} \sim \mathcal{U}(0, P^{\text{M}})$ to enable diverse task combinations that foster stable learning. Using $p^{\text{M}}$, we determine the number of masked audio tokens $n^{\text{M}}$ and uniformly sample $n^{\text{M}}$ unique indices $\mathbb{I}^{\text{M}}$ to construct a binary audio mask $\boldsymbol{M}^{\text{aud}}$ with entries in $\{0,1\}$ (0 masked, 1 unmasked) for the discrete-masking operation.

**Physical state supervision.** Building on the proposed noise-based audio masked ratio modulation strategy and adhering to REPA's representation-alignment principle (Yu et al., 2025), we introduce physical state supervision within a specific intermediate DiT block, utilizing a discrete diffusion-based objective to internalize human behavioral knowledge. First, we define that physical state tokens are generated by the SOTA pose estimator X-Pose (Yang et al., 2024), which captures essential video information regarding body gesture, facial expressions, and hand movements across $F$ video frames. Next, conditioned on the generated audio masks, we integrate a multi-layer perceptron (MLP)-based physical state projector $\text{Proj}_{\text{Phys}}(\cdot)$ into the intermediate DiT block to predict these masked physical state tokens $\boldsymbol{e}^{\text{Phys}}$ based on the unmasked audio context. The audio mask also modulates the loss, ensuring that gradients are applied only at masked positions, thus compelling the model to infer temporal dynamics from the surrounding audio input. Let $\boldsymbol{e}_j^{\text{DiT}} = \mathcal{U}_{\boldsymbol{\theta}}(\boldsymbol{x}_t, \boldsymbol{e}^{\text{ctx}}, \boldsymbol{e}^{\text{aud}}, \boldsymbol{e}^{\text{MLLM}}, t)$ denote the representation at block $j$ of a $J$-block DiT; the corresponding discrete diffusion-based physical state loss at the $J_{\text{Phys}}$-th block is given by:

$$\mathcal{L}_{\text{Discrete}} = \mathbf{1}[\boldsymbol{M}^{\text{aud}} = 0] \left\| \text{Proj}_{\text{Phys}}(\boldsymbol{e}_{J_{\text{Phys}}}^{\text{DiT}}) - \boldsymbol{e}^{\text{Phys}} \right\|^2. \quad (4)$$

## 3.4 MLLM-BASED PHYSICAL PLANNING GUIDANCE

Despite offering physical state supervision, video DiT models still lack high-level, physically grounded behavioral priors. To tackle this issue, we leverage the SOTA audio-visual omni MLLM, *i.e.* Qwen2.5-Omni Thinker, which enables joint analysis of audio and visual inputs and the planning of future states, thus offering high-level semantic guidance for avatar behaviors. We subsequently integrate MLLM embeddings into each VACE transformer block through cross-attention, effectively refining the context tokens $\boldsymbol{e}^{\text{ctx}}$, derived from reference images and frame masks $\text{M}^{\text{ctx}} \in \{0,1\}^F$ that indicate generative frame positions. This integration employs an MLP-based MLLM projector $\text{Proj}_{\text{MLLM}}(\cdot)$ that maps the combined continuous MLLM embeddings $\boldsymbol{e}_{\text{con}}^{\text{MLLM}} \in \mathbb{R}^{L^{\text{txt}} \times C^{\text{MLLM}}}$ and tokenized MLLM embeddings $\boldsymbol{e}_{\text{tok}}^{\text{MLLM}} \in \mathbb{R}^{L^{\text{txt}} \times C^{\text{MLLM}}}$ to the video latent dimension $C$, resulting in the final MLLM-guided embeddings $\boldsymbol{e}^{\text{MLLM}} \in \mathbb{R}^{L^{\text{txt}} \times C}$:

$$\boldsymbol{e}^{\text{MLLM}} = \text{Proj}_{\text{MLLM}}(\boldsymbol{e}_{\text{con}}^{\text{MLLM}} + \boldsymbol{e}_{\text{tok}}^{\text{MLLM}}). \quad (5)$$

where $C^{\text{MLLM}}$ is the MLLM channel dimension. The integration of continuous and tokenized MLLM embeddings enhances the granularity of context and reduces ambiguities arising from semantically similar embeddings (Xu et al., 2025), thereby improving guidance accuracy. Additionally, the MLLM output provides an interpretable intermediate representation of the generated video.

### 3.5 OBJECTIVE FUNCTION AND LONG-FORM VIDEO INFERENCE STRATEGY

**Objective function.** During training, we optimize our PhysAvatar via (i) a discrete diffusion-based physical state loss applied at an intermediate $J_{\text{Pose}}$-th block and (ii) a continuous diffusion-based visual loss applied at the final $J$-th DiT block. The continuous and overall objectives are:

$$\mathcal{L}_{\text{Continuous}} = \|e_J^{\text{DiT}} - v_t\|^2, \quad \text{and} \tag{6}$$

$$\mathcal{L} = \mathcal{L}_{\text{Continuous}} + \lambda_{\text{Discrete}}\mathcal{L}_{\text{Discrete}}, \tag{7}$$

where $\lambda_{\text{Discrete}}$ balances the two terms; we set it to 10 to ensure their consistent magnitude.

**Efficient long-form inference with interleaved video interpolation.** The inference process aligns with the standard video DiT model, with no audio tokens being masked. To enable long-form video avatar generation while preserving identity, we utilize the inertia of human movement—specifically, the persistence of speaker-specific gestural patterns during speech—and implement an interleaved interpolation-based inference strategy, as shown in Figure 3.

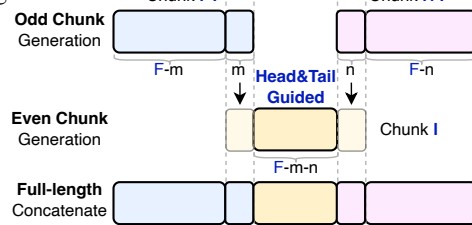

Given long audio tokens $a \in \mathbb{R}^{F^{\text{long}} \times C^{\text{aud}}}$ with $F^{\text{long}} > F$, we partition them into contiguous chunks of length $L^{\text{Chunk}}$. First, we generate all odd video chunks in parallel (non-overlapping). We then generate the even chunks via VACE frame interpolation, conditioning on adjacent odd chunks to enforce inter-chunk continuity. Concretely, each even chunk $V_l$ is conditioned on $m$ tail frames $V_{l-1}^{[F-m:F]}$ from the preceding odd chunk and $n$ head frames $V_{l+1}^{[1:n]}$ from the subsequent

Figure 3: **Illustration of the proposed inference strategy using three example chunks.**

odd chunk, where the superscripts indicate frame ranges. Finally, we iteratively concatenate the generated odd and even chunks to create the full-length video. With sufficient memory, our inference strategy permits *a minimum of two passes*—first generating the odd chunks followed by the even chunks—enabling parallel processing of non-overlapping chunks for improved efficiency.

## 4 EXPERIMENT

### 4.1 EXPERIMENTAL SETUPS

**Datasets.** We curated $\sim 200$ hours of video for training. Evaluation uses three disjoint test sets: 15 identities from our in-house dataset, 15 from PATS dataset (Ahuja et al., 2020), and 12 from the Vlogger project (Corona et al., 2025). PATS typically exhibits smaller subject-to-frame ratios, whereas Vlogger occupies a larger fraction, with both datasets predominantly featuring simple background. In contrast, our in-house test set offers greater subject-scale variation and more complex backgrounds. Notably, the evaluation is performed in a *zero-shot setting* with no test identities (IDs) were seen during training; further details on dataset curation are available in Appendix A.

**Implementation details.** Our PhysAvatar is built on Wan2.1-VACE-1.3B with $J = 30$ DiT blocks in the main branch and 15 in the VACE branch, with VACE-to-main additions applied in a distributed manner. All projectors (MLLM, audio, and physical state) along with audio and MLLM cross-attention module are trained in full, whereas the remaining modules are fine-tuned with Low-Rank Adaptation (LoRA) (Hu et al., 2022) with a rank of 128 and an alpha of 64. Experiments run on 32 H20 GPUs using AdamW (Loshchilov & Hutter, 2019) with a learning rate of $2 \times 10^{-5}$; training uses $512 \times 512$ resolution clips of 81 frames at 25 fps. Discrete diffusion-based physical state supervision was applied to the 15-th DiT block ($J_{\text{Phys}} = 15$) of the main branch to align instance-level body, facial, and hand embeddings from X-Pose (1024-d). During inference, we use odd–even overlaps $m = 5, n = 4$ and classifier-free guidance (Ho & Salimans, 2021) with 50 denoising steps, audio scale 2.0, and VACE context scale 1.1. Further details on physical state token extraction and projection, as well as the MLLM input prompt, are available in Appendix B and Appendix C, respectively.

**Evaluation metrics.** We quantitatively evaluate three aspects—visual, gesture, and facial quality—to enable comprehensive comparison with prior work. For visual quality, we report FID (Heusel et al., 2017) for frame-level fidelity and FVD (Unterthiner et al., 2018) for video-level temporal coherence, and use Q-Align for LLM-based VQA and aesthetic evaluation (ASE). For gesture quality, we compute FGD (Yoon et al., 2020) for gesture realism and DiV (Lee et al., 2019) for gesture variability; following Guan et al. (2025), we report Hand-C (hand landmark detection confidence) and Hand-V (variance of hand landmarks) to assess hand movement expressiveness and diversity. For

Table 1: **Quantitative comparison results across different test sets.** The abnormally superior performance metrics due to poor image quality in S2G-Diffusion and EchoMimic V2 are grayed.

| METHODS | Visual Quality | | | | Gesture Quality | | | | Facial Quality | | |
|---|---|---|---|---|---|---|---|---|---|---|---|
| | FID↓ | FVD↓ | VQA↑ | ASE↑ | FGD↓ | DIV↑ | Hand-C↑ | Hand-V↑ | Sync-C↑ | Sync-D↓ | CSIM↑ |
| IN-HOUSE DATASET | | | | | | | | | | | |
| S2G-Diffusion | 201.9 | 2426 | 2.34 | 1.79 | 2.73 | 2.76 | 0.61 | 1.27 | 3.72 | 7.32 | 0.884 |
| EchoMimic V2 | 148.6 | 2110 | 3.01 | 1.88 | 21.08 | 7.37 | 0.36 | 7.04 | 4.31 | 6.21 | 0.955 |
| HunyuanVideo-Avatar | 98.8 | 715 | 2.70 | 1.55 | 2.60 | 3.85 | 0.59 | 1.68 | 4.89 | 5.85 | 0.943 |
| FantasyTalking | 68.8 | 760 | 2.98 | 1.74 | 2.24 | 3.78 | 0.65 | 2.04 | 4.12 | 6.81 | 0.960 |
| OmniAvatar | 62.8 | 730 | 2.97 | 1.91 | 1.67 | 3.98 | 0.65 | 2.11 | 4.55 | 7.94 | 0.959 |
| EchoMimic V3 | 68.6 | 691 | 3.13 | 2.01 | 1.61 | 4.05 | 0.70 | 1.89 | 4.46 | 8.02 | 0.931 |
| PhysAvatar (**Ours**) | **62.3** | **713** | **3.29** | **2.13** | **1.57** | **4.33** | **0.72** | **2.33** | 4.76 | 6.02 | 0.955 |
| PATS DATASET | | | | | | | | | | | |
| S2G-Diffusion | 107.2 | 1301 | 2.82 | 1.92 | 0.40 | 1.60 | 0.74 | 2.07 | 3.52 | 6.75 | 0.909 |
| EchoMimic V2 | 60.6 | 1083 | 3.07 | 2.04 | 4.55 | 3.83 | 0.64 | 2.89 | **4.43** | 6.64 | 0.964 |
| HunyuanVideo-Avatar | 41.9 | 581 | 2.53 | 1.64 | 0.21 | 1.33 | 0.74 | 2.01 | 4.34 | 7.21 | 0.951 |
| FantasyTalking | 29.8 | 654 | 3.00 | 1.86 | 0.39 | 1.53 | 0.82 | 1.93 | 3.81 | 5.98 | 0.970 |
| OmniAvatar | 27.7 | 630 | 2.98 | 2.00 | 0.27 | 1.32 | 0.83 | 2.08 | 4.27 | 7.24 | 0.962 |
| EchoMimic V3 | 30.8 | 620 | 3.12 | 2.07 | 0.22 | 1.16 | 0.79 | 1.44 | 4.24 | 7.27 | 0.940 |
| PhysAvatar (**Ours**) | **26.9** | 583 | **3.28** | **2.26** | **0.19** | 1.72 | 0.83 | **2.35** | 4.34 | **5.56** | 0.953 |
| VLOGGER DATASET | | | | | | | | | | | |
| S2G-Diffusion | - | - | 2.28 | 1.83 | - | 3.22 | 0.54 | 2.01 | 2.40 | 5.89 | 0.892 |
| Vlogger | - | - | 3.48 | 2.08 | - | 1.14 | 0.86 | 1.60 | 5.39 | 7.42 | 0.960 |
| EchoMimic V2 | - | - | 3.26 | 1.97 | - | 5.66 | 0.47 | 4.84 | 5.00 | 5.97 | 0.966 |
| HunyuanVideo-Avatar | - | - | 2.57 | 1.41 | - | 1.15 | 0.79 | 1.05 | 5.14 | 6.21 | 0.954 |
| FantasyTalking | - | - | 3.18 | 1.82 | - | 1.50 | 0.86 | 0.87 | 4.61 | 5.97 | 0.968 |
| OmniAvatar | - | - | 3.25 | 2.01 | - | 1.18 | 0.86 | 1.18 | 5.43 | 6.56 | 0.965 |
| EchoMimic V3 | - | - | 3.50 | 2.19 | - | 1.33 | 0.78 | 1.59 | 5.55 | 6.77 | 0.954 |
| PhysAvatar (**Ours**) | - | - | **3.66** | **2.30** | - | **1.69** | **0.90** | 1.86 | 5.55 | 5.43 | 0.961 |

facial quality, we measure audio-lip synchronization using Sync-C and Sync-D (Chung & Zisserman, 2016), and facial fidelity with cosine similarity (CSIM) (Guan et al., 2025) via a face recognition model. The visual Turing test scoring standard is detailed in Appendix D. FID, FVD, and FGD metrics were excluded from the Vlogger dataset due to the absence of authentic video data. The top two results in the table are marked with **bold** for the best and underlined for the second-best result.

## 4.2 COMPARISON WITH SOTA METHODS

We systematically evaluate our PhysAvatar against several SOTA baselines with two categories: (i) explicit pose-intermediated methods (audio to pose to video), which include S2G-Diffusion (He et al., 2024), Vlogger (Corona et al., 2025), and EchoMimic V2 (Meng et al., 2025b), and (ii) end-to-end methods (audio to video), comprising HunyuanVideo-Avatar (Chen et al., 2025), FantasyTalking (Wang et al., 2025), OmniAvatar (Gan et al., 2025), and EchoMimic V3 (Meng et al., 2025a). To ensure a fair comparison, we employ DiffGesture (Zhu et al., 2023) to obtain audio-aligned pose sequences for EchoMimic V2. Notably, HunyuanVideo-Avatar is built upon HunyuanVideo (Kong et al., 2024), whereas FantasyTalking and OmniAvatar are built on the more advanced Wan 2.1.

**Results on in-house dataset.** Drawing on the quantitative results of our in-house dataset in Table 1, we present five observations. (i) Explicit pose-intermediated methods, *e.g.* S2G-Diffusion and EchoMimic V2, leverage audio to generate poses, which are then transformed into videos. This approach inevitably introduces cumulative errors, degrading visual quality. Notably, S2G-Diffusion overfits small datasets, resulting in poor generalization to zero-shot person IDs. (ii) In contrast, end-to-end methods generally achieve higher visual quality. Although HunyuanVideo-Avatar underperforms overall due to its weaker base model, its face-aware audio injection strategy yields relatively better audio-lip synchronization. (iii) FantasyTalking and OmniAvatar exhibit restricted motion diversity as their ID injection in the initial patchify layer constrains dynamic movement. (iv) EchoMimic V3 exhibits weak ID preservation due to the unified fusion of ID, audio, and text signals in a shared cross-attention module, which adversely affects the ID preservation. (v) By integrating discrete diffusion-based physical state supervision with MLLM physical planning guidance, our PhysAvatar generates more realistic videos and enhances physically grounded human behavior.

The qualitative results presented in Figure 4(a) corroborate these findings. Explicit pose-intermediated methods incur cumulative errors that degrade visual fidelity, as exemplified by S2G-Diffusion, which reveals significant hand artifacts. By contrast, end-to-end methods largely mitigate these artifacts, and models employing stronger base models further improve realism. However, FantasyTalking and OmniAvatar exhibit limited motion and weak audio-lip synchronization, while EchoMimic V3 shows poor ID preservation. In light of these challenges, our PhysAvatar surpasses all baselines, delivering enhanced visual quality along with more realistic facial expressions and body movements.

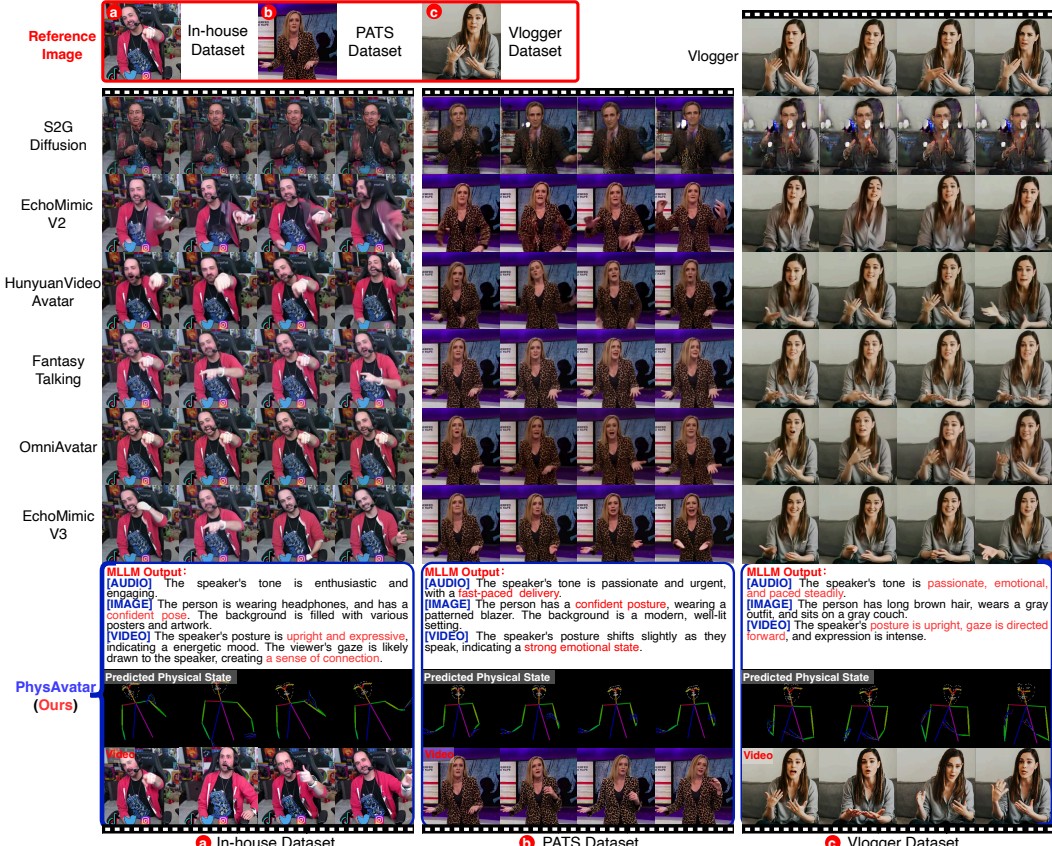

Figure 4: **Qualitative comparisons of our PhysAvatar with competing methods across datasets.**

**Results on PATS dataset.** Quantitative results of PATS dataset in Table 1 report that, despite its in-house superior performance, HunyuanVideo-Avatar exhibits marked audio-lip desynchronization when human subjects occupy only a small fraction of the frame. By contrast, our PhysAvatar remains stable under these conditions, underscoring the robustness and adaptability of our framework.

Qualitative PATS results in Figure 4(b) show that baseline methods struggle to synthesize fine-grained lip motions and handle large gesture variations, particularly when the reference image features a small mouth. By contrast, our PhysAvatar faithfully captures subtle facial dynamics and robustly handles large gesture variations while more effectively preserving subject ID.

**Results on Vlogger dataset.** Quantitative and qualitative Vlogger results in Table 1 and Figure 4(c) further confirm the effectiveness of PhysAvatar. Specifically, it outperforms all baselines, yielding richer facial detail and more expressive gesture dynamics, even in subject-dominant frames.

Figure 4 also hightlights that our PhysAvatar generates additional MLLM outputs beyond video production, thereby enhancing interpretability. The visualization of predicted physical state tokens, re-inputted through the X-Pose during the final denoising step, demonstrates strong behavioral consistency with the corresponding generated video, illustrating the effectiveness of our physical state supervision; refer to Appendix E for more qualitative results. However, the parsed hand outputs remain suboptimal due to inherent limitations and restricted parameter count of the base model.

**Visual Turing test.** Figure 1(e) displays the visual Turing test results from 15 participants evaluating our PhysAvatar against competing methods. Participants assessed five dimensions—gesture plausibility, expression appropriateness, visual quality, identity consistency, and lip synchronization—on a 1–5 scale with 0.5-point increments. The results show that our PhysAvatar achieves superior performance, particularly in gesture plausibility and expression appropriateness, confirming its effectiveness.

### 4.3 ABLATION STUDY

**Ablation on physical state supervision.** Quantitative ablation results in Table 2 show that incorporating physical state supervision consistently increases gesture quality. And qualitative results

Table 2: **Quantitative Ablation results of our PhysAvatar on the in-house dataset.**

| VARIENTS | *Visual Quality* | | | | *Gesture Quality* | | | | *Facial Quality* | | |
|---|---|---|---|---|---|---|---|---|---|---|---|
| | FID↓ | FVD↓ | VQA↑ | ASE↑ | FGD↓ | DIV↑ | Hand-C↑ | Hand-V↑ | Sync-C↑ | Sync-D↓ | CSIM↑ |
| Only Continuous | 66.6 | 746 | 3.16 | 2.01 | 2.68 | 3.67 | 0.69 | 1.24 | 4.55 | 6.23 | **0.978** |
| $J_{\text{Phys}} = 10$ (Discrete) | 64.8 | 732 | 3.12 | 2.08 | 1.70 | 4.31 | 0.68 | 2.27 | 4.65 | 6.12 | 0.956 |
| $J_{\text{Phys}} = 15$ (Discrete) | **62.3** | **713** | **3.29** | **2.13** | **1.57** | **4.33** | **0.72** | **2.33** | **4.76** | **6.02** | 0.955 |
| $J_{\text{Phys}} = 15$ ( Full ) | 66.6 | 762 | 3.15 | 2.01 | 2.60 | 4.22 | 0.66 | 2.14 | 4.40 | 6.29 | 0.956 |
| $J_{\text{Phys}} = 20$ (Discrete) | 66.9 | 752 | 3.16 | 2.03 | 2.15 | 4.08 | 0.67 | 2.20 | 4.46 | 6.31 | 0.975 |
| w/o MLLM | 67.1 | 730 | 3.10 | 1.95 | 1.74 | 3.82 | 0.71 | 1.44 | 4.36 | 6.19 | **0.958** |
| MLLM via T5 | 66.1 | 749 | 3.17 | 2.03 | 1.64 | 3.96 | 0.68 | 1.73 | 4.51 | **6.00** | 0.954 |
| MLLM via MLP | **62.3** | **713** | **3.29** | **2.13** | **1.57** | **4.33** | **0.72** | **2.33** | **4.76** | 6.02 | 0.955 |
| vanilla AR | 64.8 | 748 | 3.21 | 2.10 | 1.85 | 4.27 | 0.70 | 2.02 | 4.61 | 6.04 | 0.948 |
| Our Inference | **62.3** | **713** | **3.29** | **2.13** | **1.57** | **4.33** | **0.72** | **2.33** | **4.76** | **6.02** | **0.955** |

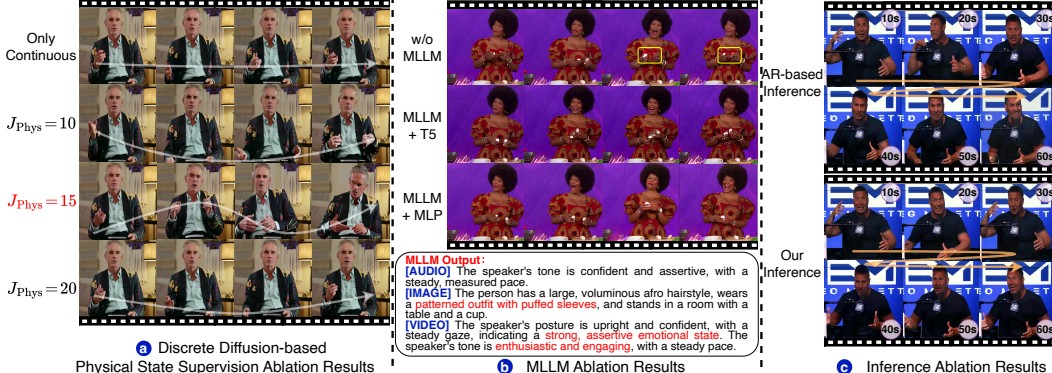

Figure 5: **Qualitative ablation results of our PhysAvatar on the in-house dataset.**

in Figure 5(a) further reveals that extreme $J_{\text{Phys}}$ values induce undesirable trade-offs and weaken gesture–visual alignment: *e.g.* $J_{\text{Phys}} = 10$ leads to exaggerated motion with blurred fingers, whereas $J_{\text{Phys}} = 20$ yields clear imagery but diminishes motion. By contrast, $J_{\text{Phys}} = 15$ achieves a favorable balance—delivering larger gesture dynamics in the first 15 blocks while emphasizing gesture–visual coupling in subsequent blocks. However, we find that full physical state supervision can negatively impact certain metrics due to individual gesture variability, even with identical audio content, highlighting the effectiveness of our discrete diffusion strategy in capturing behavioral changes.

**Ablation on MLLM guidance.** Table 2 and Figure 5(b) collectively illustrate the benefits of integrating MLLMs. Quantitative results show that MLLM integration enhances performance across nearly all evaluation metrics, with our MLP variant consistently surpassing its T5 counterpart. Qualitatively, MLLMs enable a better understanding of human-environment interactions by accurately identifying clothing patterns and minimizing texture misalignment, *e.g.* avoiding the placement of red patterns on hands. Furthermore, subjects exhibit a wider range of facial expressions with MLP integration, unlike the T5 variant, which primarily associates happiness with simplistic smiles.

**Ablation on the proposed inference strategy.** Quantitative and qualitative results in Table 2 and Figure 5(c) both highlight that the proposed inference strategy improve ID preservation and temporal coherence while maintaining gesture diversity, outperforming conventional single-direction auto-regressive (AR) motion inference. See Appendix F for a detailed discussion of our PhysAvatar.

## 5 CONCLUSIONS

In this paper, we present PhysAvatar, a novel physically grounded DiT model that generates realistic, contextually coherent, long-form avatars exhibiting human-like behavioral fidelity. By integrating discrete diffusion-based physical state supervision with multimodal language model-based physical planning guidance, we largely enhanced traditional DiT avatar training methodologies, enabling avatars to display authentic human behaviors. Our interleaved interpolation-based inference strategy further improves temporal coherence and identity preservation in long-form video generation. Experimental results demonstrate that PhysAvatar not only surpasses existing approaches in generative quality but also excels in behavioral realism across different datasets. Moving forward, we aim to enhance generative performance through the improved base model and richer datasets while optimizing real-time applications for a balance between computational efficiency and visual fidelity.

ETHICS STATEMENT

We acknowledge and adhere to the ICLR Code of Ethics throughout all aspects of our research. Our work does not involve studies with human subjects or data derived from sensitive personal information, thereby minimizing ethical concerns related to privacy and consent. We strive to ensure fairness and mitigate bias in our methodologies, especially in experimental design and data analysis.

REPRODUCIBILITY STATEMENT

This work is grounded in the VACE and VideoX-Fun[1] frameworks. To enhance reproducibility, we include comprehensive documentation of our experimental setup, along with detailed descriptions of the specific modules and the data curation process in both the main text and appendix. Furthermore, we will organize the associated training and inference code and ensure its public accessibility, facilitating the replication of our findings by the research community.

LLM USAGE STATEMENT

We employed the LLM exclusively for language refinement, including grammar correction and stylistic enhancement, to improve readability. All research ideas, methodologies, experiments, analyses, and conclusions are the exclusive work of the authors.

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

# Appendix of Paper:
# " Physically Grounded Avatar Generation"

This appendix presents essential elements to enhance understanding of our PhysAvatar: the data curation process detailed in Appendix A, the physical state token extraction and projection in Appendix B, additional analysis related to MLLM in Appendix C, detailed information about the visual Turing test scoring standard in Appendix D, more qualitative results presented in Appendix E, and a discussion of the proposed PhysAvatar in Appendix F.

## A  DATASET CURATION

Our dataset builds on a curated subset of AVSpeech (Ephrat et al., 2018) and is further enriched with high-quality, self-collected videos covering diverse scenarios such as speeches, interviews, and news reports. To guarantee consistency and reliability for downstream training, we design a rigorous multi-stage filtering pipeline to exclude low-quality or unsuitable samples.

Specifically, we remove clips with abrupt scene transitions using PySceneDetect (Castellano, 2025), as discontinuities disrupt video temporal coherence. Speaker identity is verified with Insight-Face (DeepInsight, 2025), and only single-speaker clips are retained to avoid complications from speaker alternation. To ensure precise audio–visual alignment, we further discard clips with poor audio–lip synchronization using SyncNet (Chung & Zisserman, 2016). The filtered clips are then standardized into uniform segments: single-person, upper-body shots that focus on the most informative regions—facial expressions, lip movements, and upper-body gestures. Each video is cropped and resized to $512 \times 512$ resolution, 25 fps, and a duration of 3–15 seconds.

The dataset consists of approximately 200 hours of meticulously curated high-quality audio–visual content. Its scale, diversity, and standardization make it a robust benchmark for advancing audio-driven avatar video generation. To accelerate training, we employ an offline feature extraction pipeline. Whisper is used to derive audio tokens that capture speech content and prosody, while X-Pose provides detailed physical state tokens spanning the body, face, and hands.

## B  PHYSICAL STATE TOKENS EXTRACTION AND PROJECTION

**Physical state tokens extraction.**   To integrate human behavioral dynamics into PhysAvatar, we employ the SOTA pose estimator X-Pose (Yang et al., 2024) to extract essential physical state information, including body, facial, and hand movements. As illustrated in Figure S1, X-Pose is an end-to-end multimodal pose estimation framework that can accurately detect **any** keypoints in complex real-world scenarios. It accepts an image or video clip as input and processes it through an encoder and an enhancer to improve feature extraction by leveraging information from various modalities. Subsequently, it employs two different levels of decoders, *i.e.* the **object-level** and **keypoint-level** decoders, to generate the final keypoints. The body, face, and hand keypoints are represented by the numbers $17$, $68$, and $21$, respectively, with a detailed description in Figure S2.

In this context, we define physical state tokens using the output tokens from the object-level decoder. Specifically, we select the top-$k$ tokens after the object-level decoding process: the top-1 token for both body and face, along with the top-2 tokens for each hand. Given that each token is 256-dimensional, these selections constitute the final 1024-dimensional physical state tokens for each video frame. Notably, we can input predicted physical state tokens along with their keypoint descriptions into the keypoint-level decoder during the denoising process to achieve visualization.

**Physical state token projection.**   In this work, we introduce a physical state projector that utilizes a lightweight MLP architecture designed for translating each video latent into four corresponding physical state tokens while maintaining adherence to the temporal compression ratio established by the VAE. For example, in the $f = 21$-latent configuration of the VACE framework, consistent with other models in the Wan series, the first latent token represents the initial frame. The subsequent latents are derived by compressing four consecutive frames from a total of $F = 1 + 4 \times 20$ video frames, resulting in $F = 81$. To facilitate the discrete loss calculation, we replicate the physical state token from the first frame four times, resulting in a total of $84$ *reference physical state tokens*. During

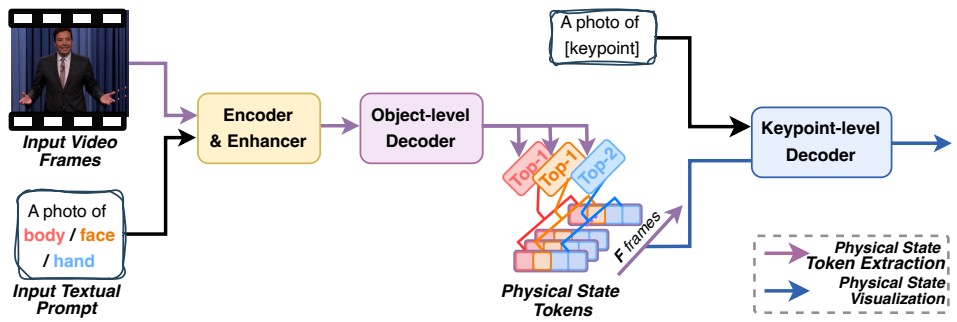

Figure S1: **Illustration of pose token extraction and visualization.**

**Body (person) :** [nose, left eye, right eye, left ear, right ear, left shoulder, right shoulder, left elbow, right elbow, left wrist, right wrist, left hip, right hip, left knee, right knee, left ankle, right ankle]

**Face :** [right cheekbone 1, right cheekbone 2, right cheek 1, right cheek 2, right cheek 3, right cheek 4, right cheek 5, right chin, chin center, left chin, left cheek 5, left cheek 4, left cheek 3, left cheek 2, left cheek 1, left cheekbone 2, left cheekbone 1, right eyebrow 1, right eyebrow 2, right eyebrow 3, right eyebrow 4, right eyebrow 5, left eyebrow 1, left eyebrow 2, left eyebrow 3, left eyebrow 4, left eyebrow 5, nasal bridge 1, nasal bridge 2, nasal bridge 3, nasal bridge 4, right nasal wing 1, right nasal wing 2, nasal wing center, left nasal wing 1, left nasal wing 2, right eye eye corner 1, right eye upper eyelid 1, right eye upper eyelid 2, right eye eye corner 2, right eye lower eyelid 2, right eye lower eyelid 1, left eye eye corner 1, left eye upper eyelid 2, left eye upper eyelid 1, left eye eye corner 2, left eye lower eyelid 2, left eye lower eyelid 1, right mouth corner, upper lip outer edge 1, upper lip outer edge 2, upper lip outer edge 3, upper lip outer edge 4, upper lip outer edge 5, left mouth corner, lower lip outer edge 5, lower lip outer edge 4, lower lip outer edge 3, lower lip outer edge 2, lower lip outer edge 1, upper lip inter edge 1, upper lip inter edge 2, upper lip inter edge 3, upper lip inter edge 4, upper lip inter edge 5, lower lip inter edge 3, lower lip inter edge 2, lower lip inter edge 1]

**Hand :** [wrist, thumb root, thumb's third knuckle, thumb's second knuckle, thumb's first knuckle, forefinger's root, forefinger's third knuckle, forefinger's second knuckle, forefinger's first knuckle, middle finger's root, middle finger's third knuckle, middle finger's second knuckle, middle finger's first knuckle, ring finger's root, ring finger's third knuckle, ring finger's second knuckle, ring finger's first knuckle, pinky finger's root, pinky finger's third knuckle, pinky finger's second knuckle, pinky finger's first knuckle]

Figure S2: **Detailed descriptions of keypoints for each body part.**

the discrete diffusion process involving physical state supervision, we employ a noise-based masked modulation strategy to randomly mask 21 audio tokens. Following this, we employ the physical state projector to predict the corresponding 84 physical state tokens. Together, the reference and predicted tokens form the basis for robust supervision in our discrete loss calculations, enabled by generated audio masks that specifically target masked positions.

## C    MORE MLLM ANALYSIS

**MLLM input prompt.**    Figure S3 illustrates the specific input prompt designed for the Qwen2.5-Omni Thinker. This MLLM input prompt performs two primary functions: (i) analyzing the audio input and the corresponding human image, denoted as **[AUDIO]** and **[IMAGE]** contents, respectively; and (ii) facilitating the planning of future avatar videos, represented as **[VIDEO]** content, while adhering to a limited word count constraint.

You are a creative director. **Analyze** the audio and reference image **together**. Your output must consist entirely of concise phrases and follow this fixed format:
- [**AUDIO**]: [Analyze **the speaker's vocal tone, emotional nuances, pacing, rhythm, intonation, and volume**. Limit to 3-5 concise phrases without questions.]
- [**IMAGE**]: [Describe **the person's key appearance, facial and pose expression, outfit, and environmental setting**. Limit to 3-5 concise phrases without questions.]
- [**VIDEO**]: [Focus on **how audio influences future human changes**: posture shifts, gaze alterations, and expression dynamics. **Indicate emotional states. Limit to 3-5 concise phrases without questions.**]
- **Constraints**: All content combined must not exceed 80 words. No filler words, conjunctions, or articles are allowed.

Figure S3: **Input MLLM prompt.**

**Text Editing Capability with MLLM.** Figure S4 presents examples of emotion-related text editing, indicating that the inherent text-conditioned editing capabilities are preserved even when the original T5 model is replaced with our MLLM-based guider. Specifically, to achieve precise text editing, we can simply append *the desired modifications as descriptions* at the end of the MLLM input prompt. Additionally, the generated video examples reveal that our PhysAvatar effectively produces realistic gestures and plausible facial expressions in text-edited scenarios, highlighting its versatility.

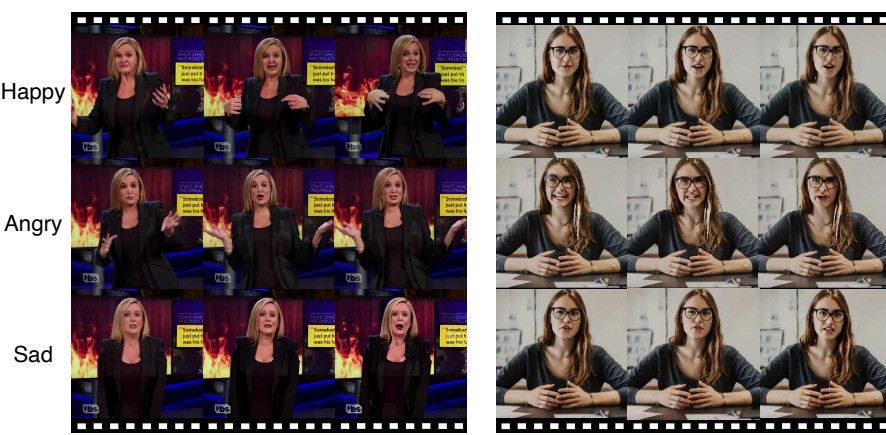

Figure S4: **Emotion-related text editing results.**

## D   VISUAL TURING TEST SCORING STANDARD

To evaluate our PhysAvatar more comprehensively beyond conventional quantitative evaluation, we conducted blinded visual Turing tests with 15 participants. They assessed 30 randomly selected avatar videos from three distinct test sets, rating each video on five key dimensions:

- **Gesture Plausibility**: Assesses the semantic compatibility of gestures with the accompanying audio. 5 – Gestures align naturally and convincingly with the audio. 4 – Gestures generally reflect the audio's intent, with minor inconsistencies; 3 – Gestures show partial or ambiguous relevance to the audio; 2 – Gestures appear unrelated or incongruent with audio; 1 – Gestures are absent, static, or clearly contradict the audio.

- **Expression Appropriateness**: Evaluates how well facial expressions convey the affective content of the audio. 5 – Expressions consistently and convincingly reflect emotional prosody; 4 – Generally appropriate with minor mismatches; 3 – Partially aligned but often ambiguous; 2 – Weak or inconsistent affective cues; 1 – Expressions absent or clearly misaligned with the audio.

- **Visual Quality**: Captures the perceived realism and rendering quality of the generated video. 5 – High perceptual realism with minimal artifacts; 4 – Visually convincing with minor imperfections; 3 – Moderate quality with visible artifacts; 2 – Noticeable degradation in texture or stability; 1 – Severe visual artifacts and low overall quality.

- **Identity Consistency**: Assesses whether the avatar's *facial* identity remains stable throughout the video. 5 – Identity is consistently preserved across all frames; 4 – Mostly stable with minor temporal variations; 3 – Occasional drift but generally recognizable; 2 – Frequent identity instability; 1 – Significant identity loss or distortion.

- **Lip Synchronization**: Assesses how accurately lip movements correspond to the audio. 5 – Precise phoneme-level alignment with natural articulation; 4 – Generally well-synchronized with minor deviations; 3 – Adequate synchronization but with noticeable timing issues; 2 – Frequent desynchronization that disrupts perception; 1 – Severe mismatch between lip motion and speech.

# E MORE QUALITATIVE RESULTS

Figure S5 presents more qualitative results that demonstrate the effectiveness of our PhysAvatar in generating realistic avatar videos exhibiting physically grounded human behavior. However, we acknowledge that some avatars still struggle to produce high-quality, fully articulated *hands*, which may exhibit artifacts during complex gestures, rapid movements, or occlusions due to the constraints of the pretrained model and the relatively limited number of model parameters employed.

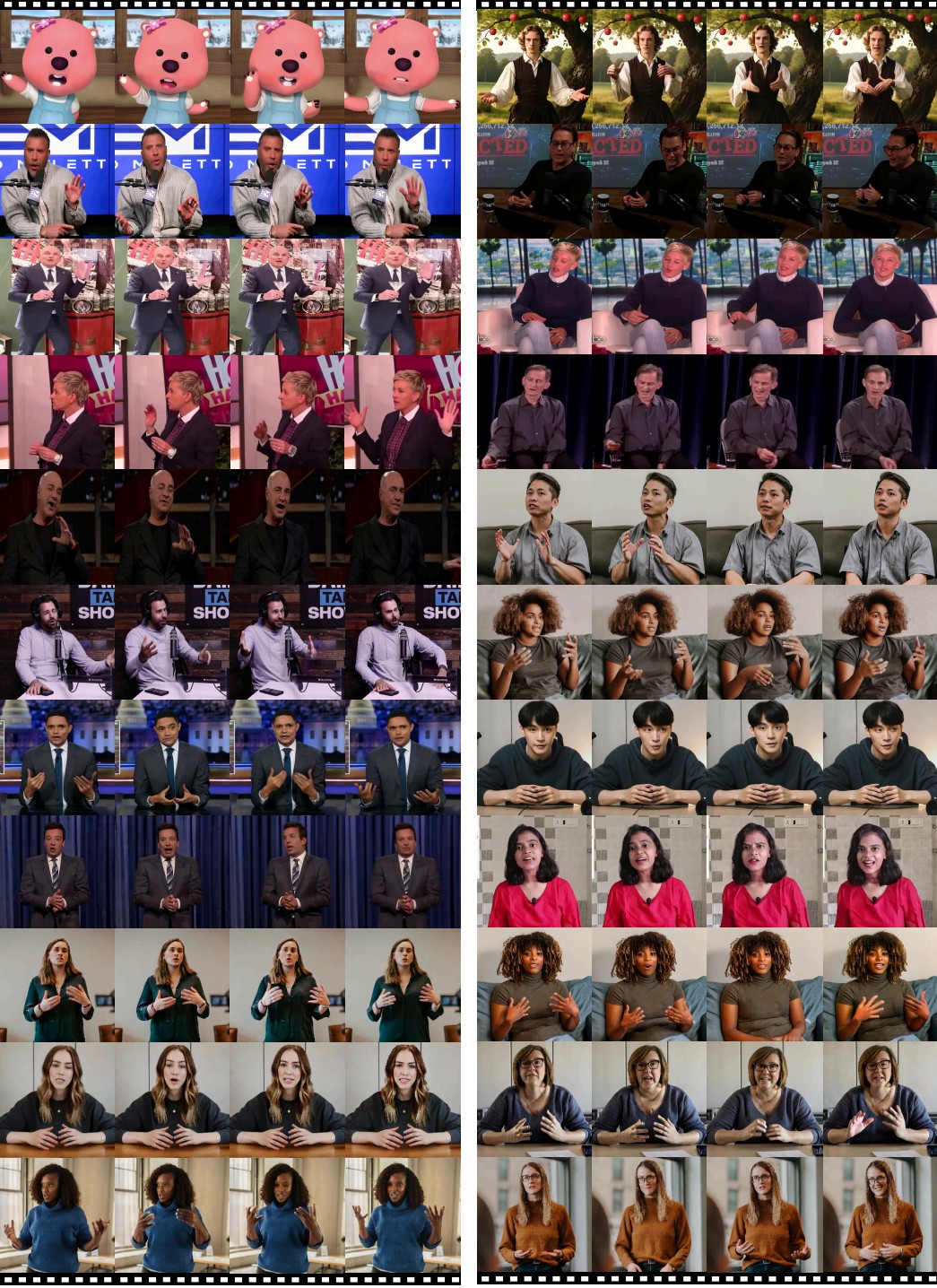

Figure S5: **More qualitative results of our PhysAvatar.**

## F    DISCUSSION

This section discusses the selection of X-Pose, the advantages and limitations of our PhysAvatar, social impact, and several aspects for future improvements.

**Discussion on the selection of X-Pose.** X-Pose provides a more comprehensive representation of human dynamics by jointly predicting both coarse-grained object information and fine-grained keypoints. This dual-level approach enables nuanced encoding of global object semantics and keypoint geometric structures through intermediate features that act as *physical state tokens*. In contrast, traditional methods like OpenPose (Cao et al., 2017) and DWPose (Yang et al., 2023) are limited to generating keypoint coordinates alone, which primarily focus on local geometry and fine-grained positioning, lacking the holistic understanding provided by X-Pose. Moreover, our preliminary experimental results indicate that the final keypoints produced by X-Pose significantly surpass those generated by OpenPose and DWPose.

**Discussion on advantages and limitations.** Our PhysAvatar framework outperforms existing SOTA baselines in both generative quality and behavioral realism, consistently producing avatars that are more physically grounded, expressive, and lifelike. However, we acknowledge a primary limitation: the quality of hand generation, which can exhibit artifacts during complex gestures, rapid movements, or occlusions due to constraints of the base VACE-1.3B model. Future iterations could benefit from utilizing larger or stronger base models to further improve the realism and fluidity of hand animations.

**Discussion on social impact.** Our PhysAvatar unlocks substantial commercial opportunities as virtual emotional companions, educational assistants, and live-streaming hosts, greatly enhancing user engagement and creating diverse revenue streams. However, these advancements also bring significant social implications. The use of advanced generative models raises concerns about misinformation, emotional manipulation, and ethical treatment of digital identities. As users form attachments to these digital entities, issues such as dependency and the erosion of real-world relationships become critical. Moreover, without proper regulatory frameworks, there is a heightened risk of exploitation or misuse. Therefore, it is essential to establish robust governance and ethical guidelines to ensure responsible deployment, balancing commercial benefits with the imperative to safeguard societal well-being and foster trust in AI technologies.

**Discussion on future improvements.** In summary, there are several key areas for improving our PhysAvatar in the future. **(i)** *Enhancements in Generative Performance.* Given the constraints of limited training datasets and computational resources, we utilized parameter-efficient methods, such as LoRA, for fine-tuning. However, we can leverage more advanced models, such as Wan2.2 or Wan2.2-based VACE, along with other robust base models, to achieve better performance. Additionally, expanding the dataset would allow for full model weight fine-tuning, further enhancing performance and fully taking advantage of the scalability of transformers Peebles & Xie (2023). **(ii)** *Adapting for Real-Time Scenarios:* Furthermore, tailoring PhysAvatar for real-time applications introduces challenges in balancing computational efficiency with visual fidelity. Future work should emphasize optimizing inference speed through methods such as Meanflow (Geng et al., 2025) or video distillation (Huang et al., 2025), as well as improving resource utilization to enhance the practicality of PhysAvatar in interactive environments.

## REFERENCES FOR APPENDIX

Zhe Cao, Tomas Simon, Shih-En Wei, and Yaser Sheikh. Realtime multi-person 2d pose estimation using part affinity fields. In *CVPR*, pp. 7291–7299, 2017.

Brandon Castellano. PySceneDetect: Python and OpenCV-based scene cut/transition detection program & library, 2025. URL https://www.scenedetect.com.

Joon Son Chung and Andrew Zisserman. Out of time: automated lip sync in the wild. In *ACCV*, pp. 251–263, 2016.

DeepInsight. InsightFace: State-of-the-art 2D and 3D face analysis project, 2025. URL https://github.com/deepinsight/insightface.

Ariel Ephrat, Inbar Mosseri, Oran Lang, Tali Dekel, Kevin Wilson, Avinatan Hassidim, William T Freeman, and Michael Rubinstein. Looking to listen at the cocktail party: A speaker-independent audio-visual model for speech separation. *arXiv:1804.03619*, 2018.

Zhengyang Geng, Mingyang Deng, Xingjian Bai, J Zico Kolter, and Kaiming He. Mean flows for one-step generative modeling. In *NIPS*, 2025.

Xun Huang, Zhengqi Li, Guande He, Mingyuan Zhou, and Eli Shechtman. Self Forcing: Bridging the train-test gap in autoregressive video diffusion. *arXiv:2506.08009*, 2025.

William Peebles and Saining Xie. Scalable diffusion models with transformers. In *ICCV*, pp. 4195–4205, 2023.

Jie Yang, Ailing Zeng, Ruimao Zhang, and Lei Zhang. X-Pose: Detecting any keypoints. In *ECCV*, pp. 249–268, 2024.

Zhendong Yang, Ailing Zeng, Chun Yuan, and Yu Li. Effective whole-body pose estimation with two-stages distillation. In *ICCV*, pp. 4210–4220, 2023.

