# OpenReview forum: "Physically Grounded Avatar Generation"
_ICLR.cc/2026/Conference — ICLR 2026 Conference Withdrawn Submission_

### Official Review · Reviewer_WkqL · 2025-10-21

**Soundness:** 3
**Presentation:** 3
**Contribution:** 3
**Rating:** 6
**Confidence:** 2

**Summary:**

This paper presents a Physically grounded DiT model for Avatar generation, PhysAvatar, It has three key contributions: (i) physical state
supervision, embedding human behavioral dynamics into the video DiT model via discrete diffusion; (ii) physical planning guidance, which leverages a multimodal language model to jointly analyze audio and visual inputs and direct the avatar behaviors according to semantic intent; and (iii) efficient long-form inference with interleaved video interpolation, improving temporal coherence and identity preservation.

**Strengths:**

1. The physical state supervision via discrete diffusion mechanism, the MLLM-based guider, the efficient long-form inference strategy improve temporal coherence and identity preservation.
2. Experimental results on in-house, PATS, and Vlogger datasets demonstrate the superior performance of the proposed model.

**Weaknesses:**

1. The approach relies on XPose and QWen. The weakness or hallucination of these models may affect the performance of the proposed method.
2. It is not clear how the proposed physical model incorporates physical signals, like force, velocity, et al.
3. The proposed approach leverages MLLM so the inference efficiency might be not very high.

**Questions:**

N.A.

---

### Official Review · Reviewer_WdnD · 2025-10-28

**Soundness:** 2
**Presentation:** 2
**Contribution:** 2
**Rating:** 2
**Confidence:** 4

**Summary:**

This paper introduces a physically grounded Diffusion Transformer (DiT) for audio-driven avatar video generation. The method integrates discrete diffusion-based physical state supervision and MLLM-guided behavioral planning to enhance realism and semantic alignment. It further employs an efficient long-form inference scheme for temporal and identity coherence. Experiments on internal and public datasets demonstrate superior performance over state-of-the-art baselines through quantitative, qualitative, and ablation studies.

**Strengths:**

1. There is a certain degree of novelty: the method leverages advanced techniques such as representation alignment (similar to REPA loss) and MLLM-guided gesture enhancement. Both quantitative and qualitative results indicate improvements over existing SOTA models.

**Weaknesses:**

1. The evaluation is conducted on a very limited test set, and the test data itself is not sufficiently challenging (e.g., reference images are low-resolution, and the scenes are relatively constrained). This makes it difficult to properly assess the performance of strong recent video-based avatar models.
2. Missing Baseline: Omnihuman-v1.5 also employs MLLMs to supplement semantic cues, yet this baseline and related discussion are missing.
3. Fine-grained gesture generation remains problematic. Although the paper provides some quantitative and qualitative analysis, the claimed gesture diversity and clarity are difficult to judge solely based on metrics and a few static frames. More convincing evidence—such as video samples—would strengthen the claims.
4. The model is trained only on approximately 200 hours video data, raising concerns about its generalization ability to broader open-domain scenarios (e.g., cartoon characters, complex backgrounds, or multilingual speech scenes).

**Questions:**

Could the authors provide more visualizations, such as comparison videos with baseline methods and additional cases that intuitively demonstrate the benefits of MLLM-guided gesture generation?

---

### Official Review · Reviewer_x3qL · 2025-10-29

**Soundness:** 3
**Presentation:** 3
**Contribution:** 3
**Rating:** 6
**Confidence:** 3

**Summary:**

This paper presents PhysAvatar, a physically grounded DiT model for avatar generation that produces realistic, contextually coherent, long-form avatars with human-like behavioral fidelity. The approach is well-motivated and shows promising results.

**Strengths:**

• Clear presentation: The paper is well-written with a logical structure that makes the technical contributions easy to follow.
• Sound methodology: The proposed framework integrating physical constraints into the DiT architecture is well-justified and addresses important limitations in existing avatar generation methods.
• Strong empirical results: Experiments on PATS and Vlogger datasets demonstrate that PhysAvatar outperforms existing baselines in generative performance.

**Weaknesses:**

1. Limited Dataset Accessibility and Reproducibility. Private datasets: Both PATS and Vlogger appear to be private/proprietary datasets, making it impossible for the community to reproduce or verify the reported results.
Suggestions:Conduct experiments on at least one public benchmark to enable fair comparison
2. No video demonstrations: For an avatar generation paper, the absence of video results is a critical omission. Static frames cannot adequately demonstrate: Temporal coherence and smoothness, Physical plausibility over time, Long-form generation quality, Behavioral fidelity claims
Suggestions: Include a comprehensive supplementary video showing: Side-by-side comparisons with all baselines (at least 3-5 examples per method, Long-form generation results (≥30 seconds) to validate "long-form" claims, Diverse scenarios: formal presentations, casual conversations, emotional expressions, dynamic gestures, Create a project page with interactive demos and downloadable results, Add at least 5-10 qualitative comparison figures in the main paper

**Questions:**

1.	Can you provide public dataset results or commit to releasing evaluation code/data?
2.	Will video demonstrations be made available?
3.	What are the failure modes of PhysAvatar?
4.	How does performance scale with sequence length?

---

### Official Review · Reviewer_yp8g · 2025-11-01

**Soundness:** 2
**Presentation:** 3
**Contribution:** 3
**Rating:** 6
**Confidence:** 3

**Summary:**

addressing the challenge of producing avatars that are both physically realistic and semantically aligned with speech. Existing methods often yield stiff or semantically inconsistent gestures. To overcome this, the authors propose PhysAvatar, a diffusion transformer framework that integrates discrete diffusion-based physical state supervision and multimodal language-guided planning. This design enables the model to capture human physical dynamics and contextual intent, generating avatars with natural motion, coherent expressions, and long-term temporal consistency.

**Strengths:**

In my opinion, the main strength of this work lies in its integration of physical realism and semantic understanding within a unified diffusion framework. Unlike prior methods that focus solely on visual or acoustic correlations, it explicitly models human physical dynamics through discrete diffusion and leverages multimodal language guidance to align gestures and expressions with the speaker’s intent.

**Weaknesses:**

The main concern is that the integration of the MLLM seems somewhat ad hoc. It works more like an additional module rather than a necessary part of the overall concept. The relationship between audio and human motion is naturally one-to-many, and adding an MLLM does not solve this problem. Instead, it makes the system more complex and increases the computational cost.

The multi-stage inference pipeline is also slow, which limits the model’s usefulness for real-time or interactive applications. The paper does not include any quantitative evaluation of inference efficiency.

In addition, errors or biases from the MLLM, especially hallucinations, may affect gesture planning and cause semantic or emotional inconsistencies in the generated avatars.

Although the paper claims to model “human physical dynamics” using discrete diffusion supervision, the “physical states” are not true physical quantities (such as forces, torques, or accelerations). They are pose tokens extracted by X-Pose, which makes the claim somewhat exaggerated.

There are no video demonstrations, making it hard to judge the visual quality and realism of the generated results.

Finally, there are serveral Typos:
Line 418: hightlights -> highlights
Table 2: VARIENTS -> VARIANTS
Figure 4: a energetic mood -> an energetic mood

**Questions:**

The semantic embedding produced by the MLLM represents global semantics rather than frame-level features, while the video diffusion model requires frame-level temporal alignment. How is the global semantic vector expanded or distributed across individual timesteps? Has the inference speed been measured, and are there any quantitative metrics provided to evaluate efficiency?

---

### Note · Authors · 2025-11-14

I have read and agree with the venue's withdrawal policy on behalf of myself and my co-authors.